# The Effectiveness of a Physical Literacy-Based Intervention for Increasing Physical Activity Levels and Improving Health Indicators in Overweight and Obese Adolescents (CAPACITES 64)

**DOI:** 10.3390/children10060956

**Published:** 2023-05-27

**Authors:** Charlie Nezondet, Joseph Gandrieau, Julien Bourrelier, Philippe Nguyen, Gautier Zunquin

**Affiliations:** 1Laboratoire Mouvement, Equilibre, Performance, Santé (MEPS), Université de Pau et des Pays de l’Adour, Campus Montaury, EA 4445, 64600 Anglet, France; charlie.nezondet@univ-pau.fr; 2L’unité de Recherche Pluridisciplinaire Sport, Santé, Société (URePSSS), Université de Lille, URL 7369, 59000 Lille, France; joseph.gandrieau@univ-cotedazur.fr; 3Laboratoire Motricité Humaine Expertise Sport Santé (LAMHESS), UPR 6312, 06000 Nice, France; 4Cognition, Action et Plasticité Sensorimotrice, INSERM UMR 1093, Université UFR STAPS Bourgogne, 21000 Dijon, France; julien.bourrelier@gmail.com; 5Departement “Unité Transversale des Activités Physiques pour la Santé” (UTAPS), Centre Hospitalier de la Côte Basque (CHCB), 64100 Bayonne, France; pnguyen@ch-cotebasque.fr

**Keywords:** intervention, physical activity, physical literacy, adolescent, body composition, cardiorespiratory fitness

## Abstract

Recently, the concept of Physical Literacy (PL) has emerged as a key concept for promoting active behavior and improving health indicators in adolescents. Overweight and obese adolescents have a low level of Physical Activity (PA), low cardiorespiratory capacity, and high Body Fat percentage (%BF). However, the development of PL in the interest of health improvement has never been studied in overweight and obese adolescents. The objective of this study was to evaluate the impact of an intervention developing PL in overweight and obese adolescents in order to increase their (PA) and improve their health. The study was a prospective, single-arm, non-randomized interventional study. The intervention brings together different actions in PA and dietary education in different adolescent living environments. The study took place over a 9-month period with two data collection times (0; +9 months) and measured Body Mass Index (BMI) and BMI z score, %BF and Skeletal Muscle Mass (%SMM), Moderate-to-Vigorous intensity Physical Activity (MVPA) by accelerometry, CRF, as well as PL by the CAPL-2 tool. Thirteen adolescents (age 11.7 (±1.09) years old) improved their PL scores (+8.3 (±9.3) pts; *p* ≤ 0.01). BMI z score (−0.3 (±0.3), *p* ≤ 0.01), their %BF (−3.8 (±4.9); *p* ≤ 0.01), their CRF (+1.5 (±1.7) mL·min·kg^−1^; *p* ≤ 0.01), and their MVPA (+4.6 (±13.7) min/day; *p* = 0.36). Initiating multidimensional interventions to develop PL in overweight and obese adolescents may be a promising prospect to enable an increase in their MVPA and improve their long-term health. Longer-term randomized controlled interventional studies are needed to confirm these findings.

## 1. Introduction

Adolescence is a critical stage in attempting to shape sustainable salutogenic health behaviors [1]. Indeed, Physical Activity (PA) behaviors in adolescence appear to determine future PA behaviors in adulthood [2]. Moreover, PA provides many benefits in adolescents: improvement of CRF, cardio-metabolic health indicators (blood pressure, fasting blood glucose…), improvement of cognitive functions and learning, and improvement of quality of life and mental health [3,4,5]. Yet, the finding of international epidemiological studies on PA levels is alarming. 80% of youths under the age of 18 are considered inactive: not meeting the PA guidelines of 60 min of Moderate to Vigorous intensity PA (MVPA) per day [6]. This chronic Physical Inactivity (PI) has serious cardiometabolic consequences for adolescents’ health [7,8].

Among all adolescents, those who are overweight and obese are vulnerable and should be a priority target for health promotion and PA. Pediatric overweight and obesity affect 26.7% of European boys and 22.9% (including obesity) of girls aged 10–19 years [9]. The main cause of pediatric obesity is an energy imbalance, which in turn is conditioned by an individual genetic risk interacting with different environmental determinants [10]. These determinants form the “obesogenic environment” [11]. PI is one of the main environmental determinants favoring the development of this pathology [12]. Indeed, overweight and obese adolescents spend an average of 22.4 min per day less than normal-weighted adolescents engaging in MVPA [13]. This PI is partly due to their high Body Fat (BF) leading to early fatigability and excessive muscle recruitment of the knee extensors [14], lower CardioRespiratory Fitness (CRF), and higher energy cost on the exercise. Lower levels of motor skills and lower physical fitness than their normal-weighted adolescents also explain the lower PA levels in overweight and obese adolescents. [13,15,16,17]. Psychological and motivational parameters such as a low sense of self-efficacy towards PA and decreased intrinsic PA motivation are also determinants explaining low PA levels [18,19,20].

Taking into account the specificities of overweight and obese adolescents regarding PA, improving the health of these adolescents through interventions that increase active behavior is, therefore, a priority. [21]. Indeed, PA has many medium- and long-term physical and psychosocial benefits for this population [22,23].PA interventions should be adapted as best as possible to target the characteristics of this population while adhering to specific PA guidelines [24].

Based on these guidelines [24], PA interventions are now recognized as providing benefits for overweight and obese adolescents. Multi-strategy interventions based on changing different behaviors (PA and diet) provide superior benefits for this population compared with a PA intervention alone [25,26,27]. These multi-strategy interventions are based on the social-ecological model, which maximizes the success of these interventions by developing actions at different levels and settings for adolescents (individual, interpersonal, environmental…) [28,29,30]. From an individual perspective, adapted exercise appears to be the most frequently mobilized lever of action in the out-of-school setting for overweight and obese adolescents [31,32]. Combined with out-of-school exercise, the school setting offers many opportunities for the implementation of actions to increase active behavior [33,34,35]. In addition to the out-of-school and school environments, household and family environments are crucial levers to help adolescents change their PA habits [33,34].

Nevertheless, the various PA support intervention for overweight and obese adolescents implemented in the literature still have heterogeneous and not always conclusive results, particularly on body composition and PA levels [26,36].

The development of new long-term intervention strategies, therefore, remains a research challenge [31] and a necessity in the clinical support of overweight and obese adolescents.

Recently, the concept of Physical Literacy (PL) [37] opened up interesting perspectives for action in favor of the health of this vulnerable public. PL is defined as “the motivation, confidence, physical competence, knowledge and understanding to value and take responsibility for engagement in physical activities for life” [38]. PL is considered a pertinent theoretical model for lifelong engagement in PA for health [39]. This model built on four interrelated domains: physical competence (physical domain), confidence-motivation (affective domain), knowledge- understanding (cognitive domain) and social participation (behavioral domain) [40] is beginning to be positively associated with health indicators [39]. An association exists between PL and MVPA [41,42] between PL and sedentary behaviors [43], between PL and CRF [44], and between PL and exercise tolerance [45]. Relationships between PL and body composition were also found, as well as a negative association with %BF and Body Mass Index (BMI) and a positive association with skeletal muscle mass (SMM) [19]. PL is also associated with motivation and self-perception toward PA [46]. PL is beginning to emerge in PA promotion interventions and is bringing the first benefits of PL development to health indicators [47,48,49]. However, an intervention based on PL development in overweight and obese adolescents has never been specifically designed. Yet, PL and its components have all the characteristics to meet the needs of this public [50]. 

This study presents the results of an intervention developing PL in overweight and obese adolescents. The main objectives were as follows:−Develop PL among adolescents;−Increase MVPA and improve health indicators (body composition and CRF) among overweight and obese adolescents.

The hypothesis is that a PA intervention based on PL development would increase MVPA and health indicators in overweight and obese adolescents.

## 2. Materials and Methods

### 2.1. Study Design and Procedure

The project CAPACITES 64 was a prospective interventional study with a single arm and was not randomized. It took place at Marracq (a secondary school in the city of Bayonne, France) in association with the Centre Hospitalier de la Côte Basque (CHCB) and its medical department “Unité Transversale des Activits Physiques pour la Santé” (UTAPS).

The intervention was organized at Marracq secondary school with the agreement of the Head of the school. The intervention as well as the evaluations were carried out by the first author of this study, a graduate in adapted exercise.

The program “CAPACITES 64” took place from September 2021 to June 2022. The different measures were collected at the baseline in September 2021 (T0) and 9 months later in June 2022 to evaluate the effects of the intervention (T1). A flowchart summarizes the study design in Figure 1.

Both the parents or legal guardians and the adolescents consented to participation and data collection, before taking part in the study. All intervention procedures were approved by the French Research Ethics Committee of the Sciences and Techniques of Physical and Sports Activities (CER STAPS n° IRB00012476-2023-20-02-233).

### 2.2. Participant Recruitment Process

Participants for this intervention were recruited from the baseline sample (4 volunteer sixth-grade classes at Marracq Middle School (*n* = 85)) found in the study by Nezondet et al. [51].

Different indicators were used to assess the health of this sample:−Anthropometric data (height, weight, BMI);−Impedance measurement (%BF and %SMM);−CRF using the 20 m walk/shuttle run test (TMNA-20) [52];−PA levels using the Youth Risk Behavior Surveillance System (YRBSS) questionnaire [53].

Following the health assessment (*n* = 85), we were able to identify adolescents who could participate in intervention on the following inclusion criteria:−Overweight or obese status (BMI > International Obesity Task Force (IOTF 25) cutoff) [54] and PA levels in among the lowest 25% (1st quartile) and/or CRF in the lowest 25% (first quartile);−A %BF in the highest 25% (4th quartile) and PA levels in the lowest 25% (1st quartile) and/or a CRF in the lowest 25% (first quartile);−Consent from the adolescent and his/her parents or legal guardians.

Non-inclusion criteria were as follows:
−Motor, mental, cognitive, or psychic disabilities in adolescents;−An injury.

Thirteen adolescents were therefore identified based on these criteria. The educational team of the school included the head of the school and physical education and sports teachers, who proposed this intervention to the adolescents and their families with an official letter.

Of the 13 adolescents included, 9 were boys (69%) and 4 girls (31%). They were 11.7 (±1.09) years old at the beginning of the intervention and 12.5 (±1.0) years old at +9 months. Eleven out of twelve variables contained the entire sample (*n* = 13), and only the MVPA variable contained eleven out of thirteen individuals.

### 2.3. Description of the Intervention CAPACITES 64

CAPACITES 64 is an educational and therapeutic intervention targeting overweight, obese, and inactive middle school students. The intervention aims to: (1) develop PL, (2) increase MVPA and (3) improve health indicators (BMI and BMI z score, %BF, %SMM and VO2peak).

Based on the social-ecological model, this intervention acts in different adolescent settings:−In school;−Out-of-school (e.g., a sports gymnasium);−In adolescents’ household environments [30].

The different modalities of the intervention were designed in accordance with national and international guidelines for PA and nutrition (Table 1) [24,55].

#### 2.3.1. Specific Actions to Develop PL and Increase PA

Recent research shows us that the use of specific actions in PA is essential for the development of PL [39,56]. Key guidelines are to develop motor skills in the context of structured and unstructured games, build strength and endurance through fun activities, and find activities that adolescents enjoy participating in. In using PA, it is fundamental to develop intrinsic motivation and confidence as well as knowledge and understanding [56].

(a)For the out-of-school activities, two weekly sessions of adapted exercise were programmed every Wednesday (3:00–5:00 pm) and Friday (5:00–6:00 pm) during the whole intervention. A total of 69 sessions were made. These sessions took place in a sports gymnasium and were supervised by an adapted exercise professional. The sessions were based on an exercise plan that respected the main criteria of adaptation, respect for the load, and progression [57].

The development of motor skills was designed using the “PLAY skills” tool from “Canadian sport for life” [58]. There are 18 motor skills classified into 5 categories: running, locomotor, object control upper body, object control lower body and balance, stability, and body control. For example: “standing and stooping”, “kicking”, “hitting with an object”, “jumping”, and “throwing over”. A typical out-of-school adapted exercise session is detailed in Table 2.

During adapted exercise sessions, several strategies have been used to develop motivation and confidence in adolescents [60,61]: −The positive influence of peers (e.g., creating a homogeneous group with similar physical characteristics);−setting short-, medium-, and long-term goals (e.g., regular review of goals);−making it possible to achieve success (e.g., adapting practice to the level of each individual);−sessions based on fun and non-structured play rather than sport competition;−positive feedback on the gain of intrinsic skills (e.g., showing the new success of an exercise).

Associated with the adapted exercise sessions, 4 education workshops were organized during the 9 months. These workshops allowed adolescents to increase their knowledge and understanding of PA.

Parents or legal guardians were invited to attend these educational workshops. These 30 min workshops covered the following topics: the definition and guidelines of PA, the benefits of PA, managing a PA schedule, and how to increase daily movements.

(b)At the school level, two main actions were implemented:

Two sessions of “health promotion” using posters provided by the library. These posters, on the theme of health promotion, were a way to bring knowledge to the adolescents. 

Health promotion action was conducted around the social sphere of adolescents; 3 awareness sessions with 7 teachers were introduced in 9 months. The three themes addressed were PA, diet, and sedentary behavior in adolescents, and the relationship between these behaviors and health. After a knowledge introduction by an adapted exercise professional, the teachers had to reflect on how to address and integrate these behaviors and information into their teaching. The teachers were able to express themselves on the actions done with the adolescents in relation to AP and dietary. The school’s actions were designed for all the teenagers in the secondary school.

(c)At the family and household level:

An initial interview and assessment were conducted at the beginning and end of the intervention between the adapted exercise professional and the family. The aim of the semi-directive interview was to know the characteristics of the adolescent and the family (availability, organization, problems) as well as the expectations in relation to the project (objectives of the adolescent…). At home, a specific environment to promote physical activity was defined in order to perpetuate the exercises and situations seen during the physical activity sessions. A practical kit was made available to all adolescents and their families: the “moving” card game (18 cards showing several illustrated stretching, strengthening, and endurance exercises with their basic number of sets and repetitions), a personalized dice (usable with the cards) and a kit of three balls. Independently, each adolescent and his or her family were given a new opportunity to perform fun exercises with individual coaching for several months. This in-home approach allows for greater involvement and understanding of the adolescent’s family.

#### 2.3.2. Specific Dietary Actions

The diet-specific actions had two objectives: to reduce weekly consumption of sugar-sweetened beverages and Ultra-Processed Foods [62,63]. 

The dietary activities were developed and carried out in accordance with the dietary recommendations for this population [64]. They were supervised by a registered dietician-nutritionist. 

Individual and collective actions, for the out-of-school, school, and home environments have been implemented. Among the main ones, we can mention:−Hands-on educational cooking workshops in the school kitchens;−Educational workshops on nutrition (similar to those on PA);−Individual consultation with the adolescent and their family based on the dietary diagnosis;−An individual follow-up on the diet of the adolescent and their family in the form of monthly exchanges (setting and reaching objectives);−Awareness-raising of the adolescent’s teachers (similar to the awareness-raising on PA).

Results regarding dietary assessments are not presented in this article because we were interested in PL and its impacts on PA and health.

### 2.4. Anthropometric Data

Height was measured to the nearest 0.5 cm using a wall height gauge (Seca^®^, Hambourg, Germany) according to the standard procedure: adolescents stood with their feet together, without shoes, leaning against a wall with head, shoulders, and feet aligned. The body mass was measured to the nearest 0.5 kg with a bioelectric impedance (BC 430 MA S TANITA, Hoogoorddreef 56E1101 Amsterdam, Netherlands). The weighing was carried out in minimal clothing (t-shirt, shorts, and socks). 

Underweight, overweight, and obese individuals were classified according to the IOTF age and gender BMI cutoffs [54]:−Underweight is characterized by a BMI < the IOTF threshold 17;−Normal weight corresponds to a BMI between the IOTF 17 and 25 thresholds;−Being overweight corresponds to a BMI between the IOTF of 25 and 30;−Obesity corresponds to a BMI > the IOTF 30 threshold and severe obesity to a BMI > the IOTF 35 threshold.

### 2.5. Body Composition Measurement

The body composition was measured with the professional bioelectrical impedance “BC 430 MA S TANITA”, Hoogoorddreef 56E1101 Amsterdam, The Netherlands). The measurement protocol was standardized and explained to each individual. The adolescent dressed in light clothing (shorts and t-shirt) without shoes or socks stood on the scale sensors. The adolescent had to remain motionless, with his or her arms outstretched during the data collection. The protocol was performed before the PA sessions. All measurements were taken on the same morning. No vigorous or intense PA was performed 12 h before the test. Adolescents had an empty bladder and had not consumed alcohol or caffeinated beverages for at least 24 h before testing. The results (rate (kg)) and %BF and SMM were then provided by the “Health monitor TANITA PRO” software (version 3.4.5).

### 2.6. Measurement of CRF by the 20 m Shuttle Walk/Run Test (TMNA-20)

The CRF was assessed by the CRF on the adapted 20 m shuttle run/walk test (TMNA-20) [52]. This test is an adapted version of the “Multistage 20-m shuttle run test” [65]. The aim of this test is to run continuously on a track with two blocks at each end, 20 m apart. The adolescent begins the test by walking at a speed of 4 km/h, then every minute the speed increases by 0.5 km/h until the adolescent stops voluntarily. The test is interrupted at the adolescent’s request or by the educator if the adolescent is no longer able to keep to the speed requested by the test tape. When the test is stopped, the Maximum Aerobic Speed reached (VMA) by the adolescent is estimated. The cardiorespiratory condition is then estimated as a correlation between the VMA (km/h) and the aerobic capacity (mL·kg·min^−1^). A mathematical formula is then used: (19.66 + (2.21 × VMA) + (0.05 × age) + (2.08 × girl (0) or boy (1)) − (0.38 × BMI)).

### 2.7. Measurement of PA Levels

PA levels were assessed and recorded using Actigraph GT3X+ accelerometers (ActiGraphTM, Pensacola, FL, USA) and Actilife software (version 6.13.4), which are valid and reliable tools for recording and measuring PA levels in adolescents [66,67]. 

Adolescents wore the accelerometer on a belt at their hips for 7–10 days. The accelerometers were set to record 30 Hz accelerations. The accelerometer data were then downloaded in 1s epochs, visually inspected, and processed with the following procedures. To validate a period of accelerometer wear, we used Troiano’s parameters [68] with a period of non-wear classified as 60 min or more of zero counts. In addition, only participants who wore the accelerometer for at least 3 days and at least 10 h per day were included in the statistics [67,69]. PA levels specific data (Cut point and MVPA) were calculated using [70]. Data above 2296 counts/min define MVPA.

### 2.8. Measurement of PL

Adolescents’ PL was measured by the Canadian Assessment of PL Second Edition (CAPL-2). This tool is one of the first comprehensive protocols to assess the various components of PL [71], he use of the CAPL-2 is valid and reliable in an adolescent population [19]. The score assessment is divided into 4 domains that when added together form a final PL score out of 100 points: Daily Behavior, Physical Competence, Motivation and Confidence, and Knowledge and Understanding. The different assessments present in the 4 domains are described in Figure 2.

According to their score, gender, and age, adolescents are classified into 4 different levels: Beginning, Progressing, Achieving, and Excelling. The “Beginning” level means that the adolescents have a limited and insufficient PL level compared to youth of the same age and gender. The “progressing” level means that the adolescents are performing at a similar level to the youth of the same age and gender (an average level). The “achieving” and “excelling” levels mean that the adolescents meet or exceed the recommended PL level [71].

### 2.9. Statistical Analysis

All statistical analyses were carried out with Statistica (version 7.1). The mean, standard deviation, median, and distribution of the data were calculated for each variable: Age, BMI, %BF, %SMM, maximal aerobic speed (Vmax km/h), VO2peak (mL·kg·min^−1^), PL score (pts), MVPA (min/day).

The normality of each variable was tested using the Shapiro-Wilk test. A *p*-value ≤ 0.05 was chosen for statistical significance.

Correlation matrices are used to determine the relationships of the different variables: BMI, %BF, %SMM, VO2peak (mL·kg·min^−1^), MVPA in baseline (T0), and post-intervention (T1). The correlation coefficient (r) will be used to quantify the strength of the linear relationship between the variables. Statistical significance is indicated by a *p*-value ≤ 0.05. 

A linear trend line was used to illustrate the correlation between PL and MVPA at baseline (T0) and after 9 months (T1).

The impact of the intervention at 9 months is indicated by gross mean and percentage data (T1–T0). The Wilcoxon nonparametric rank test for matched groups was used to compare data. A *p*-value ≤ 0.05 was chosen for statistical significance. 

## 3. Results

### 3.1. Baseline Results (T0) and Impact of the Intervention at 9 Months on the Group of Adolescents (T1)

Baseline characteristics at pre-intervention (T0) and 9 months (T1) showing the impact of the intervention are presented in Table 3. The following sample had a mean age of 11.7 (±1.09) years old at T0 and 12.5 (±1.0) years old at +9 months.

Between T0 and T1 the total PL score increased by 8.3 (±9.3) points (51.5 to 59.8) equivalent to 16% (*p* ≤ 0.01). If we analyze in detail each PL domain assessed by the CAPL-2 tool, we find a significant positive impact of the intervention for the domains of “physical competence” + 4.1 (±4.1) points equivalent to 35.5% improvement (*p* ≤ 0.01) and “knowledge and understanding” + 2.5 (±2.3) points equivalent to 45% improvement (*p* ≤ 0.01). The domains of “daily behavior” and “motivation and confidence” were also improved with the intervention but not significantly (respectively: +1.0 (±5.9) points equivalent to 12.8% improvement (*p* = 0.5) and +0.7 (±3.8) point equivalent to 2.9% improvement (*p* = 0.7)).

The development of PL was expected to improve health indicators and increase MVPA. Regarding health indicators, the 9 months of intervention significantly increased Vmax by 0.5 km/h (±0.7) equivalent to 5.7% (*p* = 0.03), and VO2peak by 1.5 mL·min·kg^−1^ (±1.7) equivalent to 4.8% (*p* ≤ 0.01) in the adolescents group. Anthropometric and body composition data were also improved by the intervention, %BF decreased by 3.8% (±4.9) (*p* ≤ 0.01). %SMM increased by 2.2% (±2.8) (*p* ≤ 0.01).

BMI showed a small non-significant decrease −0.9 (±1.5) (*p* = 0.07) equivalent to a 3.7% loss. However, the BMI z score decreased significantly by −0.3 (±0.3) (*p* ≤ 0.01) equivalent to a 16.4% decrease. MVPA increased by 4.6 min/day (±13.7) equivalent to 8.5% for adolescents but not significantly (*p* = 0.36).

### 3.2. Correlation Matrix between Variables at T0 and T1

A correlation matrix presents the associations between the different variables at baseline (T0) and at 9 months post-intervention (T1) (Table 4).

During the different data collection periods, VO2peak shows a significant positive correlation with %SMM (*p* ≤ 0.05) and a significant negative correlation with %BF (*p* ≤ 0.05). This demonstrates that VO2peak is strongly associated with body composition data (%SMM and %BF) and that when VO2peak increases %SMM increases and %BF decreases. 

The relationship between PL score and VO2peak is also positive at T0 and T1 (*p* ≤ 0.01). This means that the PL score is strongly associated with VO2peak and when the PL score increases, VO2peak also increases. In contrast to baseline results (T0) (r = 0.31), the MVPA variable and VO2peak are associated at 9 months (T1) (r = 0.81; *p* ≤ 0.01). 

Regarding the correlation between PL and MVPA, the two variables were not correlated at baseline (T0) (r = 0.39). After 9 months of intervention (T1) we find a positive correlation between PL and MVPA (r = 0.76; *p* ≤ 0.01) (Figure 3). These results show that at baseline (T0) PL and MVPA were not linked to each other and that at 9 months (T1) PL and MVPA values tend to increase together.

## 4. Discussion

The aim of this study was to evaluate the effectiveness of an intervention based on the development of PL to increase MVPA and improve health indicators in overweight and obese adolescents. This is the first study to expose the effectiveness of an intervention to increase PL in overweight and obese secondary school students. Indeed, through the preliminary results at 9 months, the average PL score of the secondary school students increased by 16% (*p* ≤ 0.01). Increased PL scores resulted in a 5% (*p* ≤ 0.01) improvement in VO2peak, a 2.2% (±2.8) (*p* ≤ 0.01) increase in %SMM, and a 3.8% (±4.9) (*p* ≤ 0.01) decrease in %BF. However, increasing the mean PL level did not significantly increase MVPA (+4.6 min/day (±13.7); *p* = 0.36). 

### 4.1. Development of PL

To improve and maintain good health in adolescents over the long term, the first aim was to make them more physically active by increasing their PA levels [26]. To achieve this, many interventions that meet specific guidelines were implemented [24,72]. Yet, results on increasing PA levels and improving health indicators are still heterogeneous due to a lack of long-term follow-up as well as support that does not always target all the needs of overweight and obese adolescents [36]. 

There is a need to explore new strategies for PA intervention with this population, the integration of the new PL concept is warranted and necessary [50,73]. 

The results of this study confirm the interest of an intervention in the development of PL. Indeed, after the 9 months of intervention, the average total score of PL assessed by the CAPL-2 tool increased from 51.5 to 59.8 out of 100, an increase of 8.3 (±9.3) points (*p* ≤ 0.01). With the interpretation of the CAPL-2 total score, the group of adolescents moved from the “beginning” level (mean score < 51.9) to the “progressing” level (51.9 < mean score < 69.6). If we look at the results according to the different components of PL, we can see the most important improvements in the “physical competence” score with an increase of 4.1 (±4.1) out of 30 points (11.7 to 15.9) and a shift from the “beginning” level to the “progressing” level. The “Knowledge and understanding” component increased by 2.5 (±2.3) points out of a total score of 10 points (5.6 to 8.2), which means a shift from the “progressing” level to the “successful” level. Smaller improvements in “daily behavior” component (+1 (±5.9) point and maintained in “progressing” level) and in “motivation and confidence” component (+0.7 (±3.8) points and maintained in “progressing” level) were found.

The different improvements obtained in the PL domains are explicable. Indeed, the domain “motivation and confidence” is evaluated through a self-reported questionnaire which leads to a complex perception of this domain by adolescents. [74]. Moreover, the high initial level of adolescents (22/30 points) may explain the low increase in this domain.

In scientific literature, results on interventions aimed at developing PL are still recent. Indeed, the study by Kwan et al. [49] on 65 students in the transition to university showed an improvement in PL scores for the intervention group but these results did not show statistical significance (*p* = 0.06). The meta-analysis by Carl et al. [47] also concludes that there is a development of PL through interventions. Conversely, in the study by Li et al. [74] carried out on elementary school children, we found a significant improvement in the PL score in its “physical competence” and “knowledge and understanding” components in both intervention groups. The comparison of our results with the scientific literature shows that the improvement of PL after an intervention is a variable according to the public and their initial level of PL. In overweight and obese adolescents, this type of intervention shows significant results both statistically and clinically. This intervention allowed to move the group from a “beginning” level meaning a limited and unsatisfactory PL level to an average “progressing” level meaning a PL level in the mean of people of their age. 

### 4.2. Effect of PL Development on Body Composition and Anthropometric Data

Body composition assessment by %BF and %SMM are markers of health in overweight and obese adolescents. We know that keeping a high SMM and low BF during childhood and adolescence is predictive of better metabolic health in adulthood [75]. Regarding PL and body composition, we knew that high PL was associated with low BF and high SMM [19]. However, no study has evaluated the impact of increased PL on body composition.

The effects of the intervention at 9 months reduced the average %BF from 30.1% to 26.3% of total body mass. The study by Ogden et al. [76] compares, between 1999 and 2004, the % of BF without intervention in children and adolescents aged between 6 and 19 years old. They find a constant evolution of the % of BF in girls between 11 and 13 years old (+0% (±0.7)). In contrast, between 11 and 13 years old, the % of BF in boys decreased slightly (−1.8% (±0.7)).

The same intervention increased the %SMM from 39.5 to 41.7% of the adolescents’ total body mass. If we take the initial average body mass (T0) of the sample which is 64.5 kg, the average BF was 20.2 kg and the average SMM was 25.1 kg. At 9 months of intervention, the mean body mass was 67 kg with a mean BF of 18.1 kg and an SMM of 27.7 kg. 

These results are in agreement with the meta-analysis by Soare et al. [72] which shows a mean reduction of 7% in BF and a mean increase of 7% in lean body mass on interventional PA studies targeted at overweight and obese 5–17 year old.

Regarding anthropometric data, this intervention did not significantly improve the adolescents’ BMI (−0.9 kg/m² (±1.5); *p* = 0.07). However, the BMI z score was significantly improved (−0.3 (±0.3) *p* = 0.003), which means that the average BMI of the adolescents was nearer to the average BMI of adolescents of their age (IOTF standards). The improvement in %SMM and decrease in %BF remains superior to the improvement in BMI in terms of metabolic health for overweight and obese adolescents [75]. Especially since the numerous meta-analyses studying the effect of PA on BMI yield heterogeneous and non-robust results [26,77].

### 4.3. Effect of PL Development on CRF

In 9 months of intervention, their mean VO2peak increased by 1.5 mL·min·kg^−1^ (±1.7) from 30.9 to 32.4 mL·min·kg^−1^. This average improvement was concretely demonstrated with the increase of one step in the TMNA-20 test (+0.5 km/h VMA) [52]. This result is confirmed by the interventional study by Kwan et al. [49] who studied the effectiveness of a program developing PL on 65 students in secondary school. PL development increased CRF in the intervention group (+1.84 mL·min·kg^−1^) in contrast to the control group (−0.24 mL·min·kg^−1^). Maintaining or increasing CRF (VO2peak) in overweight or obese adolescents is a protective health parameter [78]. Indeed, better levels of CRF are associated with cardiometabolic protective factors such as lower blood pressure as well as a better lipid profile [79,80].

A PA intervention based on the development of PL, therefore, improves CRF. In overweight and obese adolescents, the improvement in CRF is not due to an increase in maximal oxygen consumption during exercise but to a change in body composition directly implicated in the measurement of CRF [16]. Indeed, baseline and post-intervention correlational analysis show that %BF and VO2peak showed a negative association and %SMM and VO2peak showed a positive association during T0 and T1. This means that %BF and %SMM are strongly correlated with VO2peak. 

Therefore, the development of PL allows decreasing the %BF and increasing the %SMM, which have a direct impact on the improvement of the CRF measured during a practical field test.

### 4.4. Effect of PL Development on MVPA

This intervention based on PL development was expected to increase MVPA. Indeed, PA mediates the relationship between PL and health indicators [45]. Furthermore, Cairney et al. [39] stated that PA and PL were directly related due to the fact that these two components develop in a reciprocal manner. To confirm this, some interventional studies find an increase in MVPA after an intervention based on PL development. The Physical Education and Physical Literacy (PEPL) program [81] show an increase in MVPA of 5.7 min (*p* ≤ 0.05) in 318 fifth-grade students (age 10.4 years ± 0.4). Regarding this study, MVPA did not increase significantly (+4.6 min/day (±13.7); *p* = 0.36) between the two assessment times.

The non-significance of these results may be due to the characteristics of our sample, as well as an important standard deviation of our results but also to the limitations of the accelerometer used for measuring MVPA [82]. Despite the non-significance of the increase in MVPA, this result remains clinically significant because it shows that an intervention developing PL can avoid the decline in MVPA encountered during adolescence. [83]. Two studies, one on adolescents and another on university students, confirm this conclusion. When failing to significantly increase MVPA, an intervention based on the development of PL makes it possible to avoid the decline in PA behavior in adolescents [49,84]. 

The correlation matrix brings us information about the correlation between PL and MVPA. Indeed, on the baseline results (T0) we do not find any correlation between these two variables (r = 0.39). After 9 months of intervention (T1), a strong positive correlation appears between PL and MVPA (r = 0.76; *p* ≤ 0.01). This correlation showing strength between PL and MVPA data after intervention could provide an answer to the evolution of MVPA after intervention on PL development in this population.

### 4.5. Correlation between PL and MVPA

Correlation analysis between PL and MVPA at baseline (T0) and after intervention (+9 months (T1)) provides results on the evolution of the relationship between these two variables. We can see that these two variables were not correlated with T0 but were positively correlated with T1. Because of the increase in the average PL score with the intervention (+8.3 points), we can suppose that the PL variable is related to the MVPA variable only after a certain threshold of PL score. This would explain the non-correlation between PL and MVPA at T0 and its correlation at T1. This hypothesis is supported by several studies found in the literature [43,45]. Indeed, these studies show that the PL variable is correlated with MVPA with population samples with high PL scores (e.g., 63.6 (±10.5) points on the total PL score (CAPL-2)) [43].

For the intervention, the hypothesis that PL is related to MVPA above a certain threshold of PL score could partly explain the non-significantly results of the impact of the intervention on MVPA improvement (+4.6 min/day (±13.7); *p* = 0.36). Indeed, low initial PL scores in overweight and obese adolescents may have been a barrier to increasing MVPA with this intervention. At 9 months, PL development intervention does not appear to be sufficient to develop MVPA in overweight and obese adolescents with the lowest levels of PL and MVPA but may have an impact on increasing MVPA when adolescents already have some level of PL. Longer-term intervention and follow-up are needed to determine the effect of PL development on active behavior of adolescents with the lowest PL scores.

This finding may help explain the difficulty in increasing MVPA with PA interventions in overweight and obese adolescents [36,85]. Their low level of PL score [86] could inhibit the increase of MVPA. For adolescents with the lowest baseline PL levels, the development of PL during 9 months does not seem to be sufficient to increase their MVPA levels. This can be explained in several ways: the adolescents with the lowest PL levels are also the least active. These less active adolescents have more individual barriers (lower motor skills and motivation) and environmental barriers (access to PA, parents’ education, and lifestyle habits…) to increasing their MVPA than do adolescents with higher PL and MVPA levels [87,88].

### 4.6. Strengths and Limitations of the Study

This study presents strengths in the originality of its results as well as in the innovation of the accompaniment of the overweight and obese adolescents. Indeed, it is the first study to evaluate the impact of an intervention based on the development of PL in the increase of MVPA and the improvement of health in overweight and obese adolescents. This study answers one of the gaps in the literature, which is the possibility of implementing a long-term PL development intervention on a specific population [73].

Indeed, other interventional studies had investigated the impact of a PL-based intervention but only in a prevention setting with healthy university and school populations [49,74]. This study provides interventional results on the effect of a PL intervention in overweight and obese adolescents. These results add to the information from the only non-interventional study of PL and pediatric obesity. [50]. 

This study also presents a rigorous method regarding the evaluation of the different indicators. PL was assessed using one of the two most comprehensive assessment tools, the CAPL-2 tool [71]. Data related to MVPA and health indicators are collected and analyzed through reliable and valid methodologies namely accelerometer, bioelectrical impedance, and aerobic fitness test (TMNA-20) [52,67,89]. Although preliminary, these results are encouraging. Indeed, they suggest a promising strategy that PL development can increase active behaviors and improve certain health indicators in overweight and obese adolescents [81,84]. 

This study also has limitations. The first is that it is a single-arm, non-randomized study with no control group. This reduces the internal validity of the study. However, the single arm was decided for human and financial reasons. One of the perspectives to confirm its results will be to carry out a similar randomized controlled study. The second limitation is the characteristics of the sample. The sample was non-random, which presents a possible recruitment bias. In addition, a larger sample size will allow the results to be generalized to the entire overweight and obese adolescent population. The failure to control for biological parameters such as pubertal development and growth is also a limitation. These parameters directly influence adolescents’ CRF but do not influence PL levels [90,91]. The last limitation involves the accelerometer tool. Wearing it was difficult for some adolescents, which led to a loss of data. Indeed, wearing it in a social context (school) sometimes causes discomfort and hinders data collection.

## 5. Conclusions

This study suggests that initiating interventions to develop PL in overweight and obese adolescents is a promising strategy. Indeed, the first results at 9 months show an increase in PL scores, an increase in CRF and %SMM, and a decrease in %BF. In addition, the development of PL to avoid the decline of MVPA present in adolescence. This study shows that having a sufficient PL score is essential to generate an increase in MVPA. In these overweight and obese adolescents, the goal of interventions could be to increase PL and then expect an increase in PA levels. The results of this pilot study are encouraging with the objective of maintaining active behavior throughout life in this population. However, longer-term results with a larger sample size are needed to determine the maintenance of the various health benefits for adolescents and possibly a greater impact on PA levels.

## Figures and Tables

**Figure 1 children-10-00956-f001:**
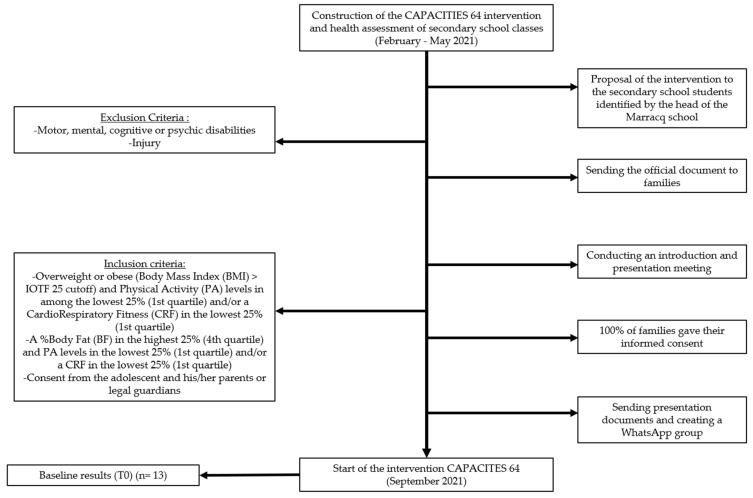
Study design of the CAPACITES 64 project.

**Figure 2 children-10-00956-f002:**
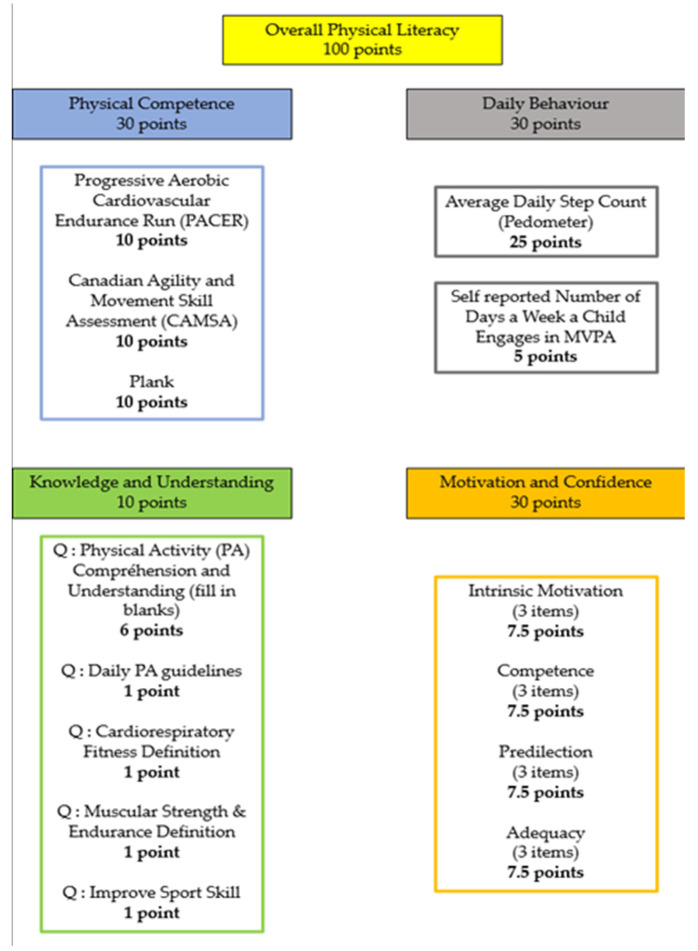
A detailed figure of the point system in PL evaluation from the instruction manual (CAPL-2).

**Figure 3 children-10-00956-f003:**
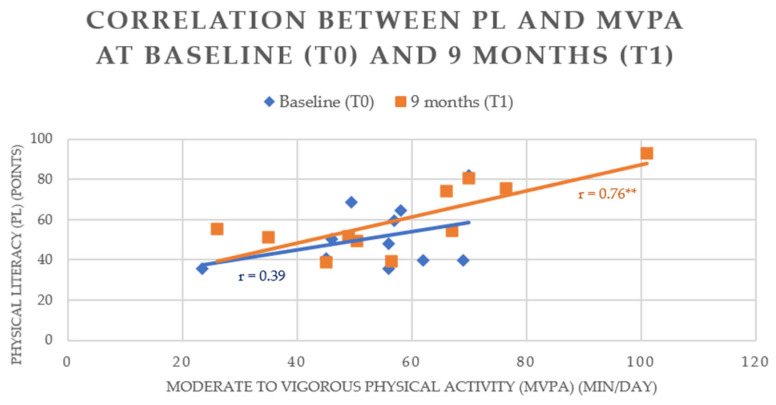
Correlation between PL and MVPA at baseline (T0) and 9 months (T1). ** *p*-value ≤ 0.01.

**Table 1 children-10-00956-t001:** Guidelines in PA for overweight and obese adolescents. The «Physical activity guidelines» table was adapted from O’Malley and Thivel [24].

Age	Type	Time	Benefits
5 to 17 years old	− Moderate-to-Vigorous intensity Physical Activity (MVPA) including activities to promote bone health (jumping, running…) − active transportation, organized and nonorganized PA, games, physical education, and other activities at home, school, work, and in the community.	− At least 60 min per day − At least 3 times/ week	− Promote concentration and learning − Build bone and muscles − Improve movement skills and coordination − Improve balance − Maintain body mass and improve health − Encourage self-confidence and develop social skills − Improve mental health and well-being

MVPA: Moderate-to-Vigorous Physical Activity; PA: Physical Activity.

**Table 2 children-10-00956-t002:** Organization of a typical session.

Organization of a Adapted Exercise Session	
Warm-up	Modalities: Time: 5 to 10 min—intensity between 50 and 70% maximum Heart Rate (HR max) Construction: 100% in the form of games with intensity progression (from 50% to 70% HRmax). Guidance according to work in the body of the session (aerobic, muscular…)
Session	Modalities: Time: 40 to 50 min—Intensity: between 50 and 90% HR max. Principle of intermittent exercise with high periods at 90% HRmax and low periods at 50% HRmax. The work at the target HR zones is controlled with a heart rate monitor. Construction: − 60% (30 min) in the form of traditional games (“pass to 10”, “rabbit hunter”) or structured activities (“handball”, “hockey…”) with an aerobic focus. Change in the practice environment: outdoor, indoor, in the air − 40% (15 min) in the form of an exercise circuit to develop more specific motor skills, to work on muscle strengthening, coordination, or balance. For the exercise circuit, the intensity is 5–6 on the Rating of Perceived Exertion (RPE) scale (Borg CR10) [59]
Rest	Modalities: 5 to 10 min Construction: Time for exchange on the session: perception, feelings, and transmission of information

Min: minute; HR max: maximum Heart Rate.

**Table 3 children-10-00956-t003:** Baseline characteristics of adolescents (mean ± standard deviation) and impact of 9 months of intervention (T1).

Characteristics (*n* = 13)	Baseline (T0)	Post Intervention 9 Months (T1)	Gross Difference	Difference in %	*p*-Value (*p* ≤ 0.05)
BMI (kg/m²)	26.4 (±5.5)	25.5 (±5.1)	−0.9 (±1.5)	96.7	0.07
BMI z score	2.0 (±0.8)	1.7 (±0.9)	−0.3 (±0.3)	83.6	≤0.01 **
SMM (%)	39.5 (±4.9)	41.7 (±4.4)	2.2 (±2.8)	105.4	≤0.01 **
BF (%)	30.1 (±8.7)	26.3 (±7.8)	−3.8 (±4.9)	87.4	≤0.01 **
Vmax (km/h)	8.7 (±1.5)	9.2 (±1.6)	0.5 (±0.7)	105.7	0.03 *
VO2peak (mL·min·kg^−1^)	30.9 (±4.4)	32.4 (±4.6)	1.5 (±1.7)	104.8	≤0.01 **
MVPA (min/day)	53.8 (±12.9)	58.4 (20.8)	4.6 (±13.7)	108.5	0.4
PL (CAPL-2)					
Total score PL	51.5 (±14.1)	59.8 (±18.2)	8.3 (±9.3)	116.2	≤0.01 **
Score in the «physical competence»	11.7 (±7.9)	15.9 (±7.8)	4.1 (±4.1)	135.5	≤0.01 **
Score in the «daily behavior»	11.9 (±5.0)	12.9 (±7.0)	1.0 (±5.9)	112.8	0.5
Score in the «knowledge and understanding»	5.6 (±2.0)	8.2 (±1.9)	2.5 (±2.3)	145.0	≤0.01 *
Score in the «motivation and confidence»	22.2 (±4.8)	22.9 (±4.6)	0.7 (±3.8)	102.9	0.7

BMI: Body Mass Index; SMM: Skeletal Muscle Mass; BF: Body Fat; Vmax: Maximum aerobic speed; VO2peak: maximum oxygen volume; MVPA: Moderate to Vigorous-intensity Physical Activity; PL: Physical Literacy; * *p*-value ≤ 0.05; ** *p*-value ≤ 0.01.

**Table 4 children-10-00956-t004:** Correlation matrices between baseline (T0) and 9-month post-intervention (T1) data collection between PL, body composition data, CRF, and MVPA.

Baseline T0	PL	MVPA	VO2peak	%BF	%SMM
PL (points)		0.39	0.83 **	−0.41	0.41
MVPA (min/day)	0.39		0.31	−0.16	0.16
VO2peak (mL·min·kg^−1^)	0.83 **	0.31		−0.72 *	0.72 *
%BF	−0.41	−0.16	−0.72 *		−0.99 **
%SMM	0.41	0.16	0.72 *	−0.99 **	
T1	PL	MVPA	VO2pic	%BF	%SMM
PL (points)		0.76 **	0.91 **	−0.37	0.37
MVPA (min/day)	0.76 **		0.81 **	−0.33	0.33
VO2peak (mL·min·kg^−1^)	0.91 **	0.81 **		−0.61 *	0.61 *
%BF	−0.37	−0.33	−0.61 *		−1.00 **
%SMM	0.37	0.33	0.61 *	−1.00 **	

SMM: Skeletal Muscle Mass; BF: Body Fat; VO2peak: maximum oxygen volume; MVPA: Moderate to Vigorous-intensity Physical Activity; PL: Physical Literacy. * *p* value ≤ 0.05; ** *p* value ≤ 0.01.

## Data Availability

The datasets used during the current study are available from the corresponding author upon reasonable request.

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
