# Peer review of "The Effectiveness of a Physical Literacy-Based Intervention for Increasing Physical Activity Levels and Improving Health Indicators in Overweight and Obese Adolescents (CAPACITES 64)"

_children, 2023, doi:10.3390/children10060956_

Round 1

Reviewer 1 Report

Line 93. Where it says an should read a.

Participants. I believe that the low number of participants joining the study is a major shortcoming.

What is the reason why not all participants are part of the MVPA-related measurement?

Description of the intervention. How do you ensure that participants work at their target heart rate?

Lines 189-190. I think this is a handicap and not a strength. What happens to the students who do not fit into the different groups according to the criteria of goals to be achieved?

Line 194. How is this monitored?

Line 200. Are all the students involved, and their families? I think this is something that could be solved by incorporating the information into the sessions themselves, both in the school and out-of-school context.

How is the intervention monitored at household level together with families?

Line 229. How is this intervention monitored?

Is it possible to know about the effectiveness of diet-related measures?

Results. What could explain the lack of behavioural change if they are attending an intervention for 2 sessions and 3 hours per week?

Results. This is probably the most important assessment to pay attention to.
The intervention, among others, aims to develop PLand MVPA. So how is it possible to know that PL influences MVPA? Is it possible to isolate this effect or am I not being able to interpret it because of the English language? If this is not possible, the title should be changed and the whole article should be revised.

How can it be explained that PL increases variables associated with PA practice but not MVPA?

Line 432. The references are associated more with PA than with PL. There seems to be no clear evidence regarding the influence that PL may have in relation to body composition variables and anthropometric data.

Line 461. I still do not understand why the improvement in CRF is associated with PL and not with other aspects that may be mediating it. In fact, it is indicated that the improvement is due to the change in body composition which, in turn, may not be due to the growth process of the participants? How is it possible to clearly say that the improvement in PL produces, in turn, improvements in CRF?

I believe that a control group would have made it possible to add robustness to the researchers' findings.

I have no comments related to this aspect.

Reviewer 2 Report

Your paper provides very many information. On the other hand it is very difficult to understand your main goals. I would like you to consider to rewrite the method section and describe your participants more precise and as well as the study more accurate.

In addition, I suggest to describe the whole project in the introduction. At the beginning of the method section, it was a little surprise. 

The English language is fine to me. I suggest a native speaker should do minor changes. 

Reviewer 3 Report

Dear authors. The subject of this study is relevant, however, the presentation of the manuscript must be improved. I think that the current study could be presented as two manuscripts. One would be about the presentation/validation of the project. The other would be with the results. These are suggestions to be evaluated by all the authors and by the Editor of this journal. Considering the evaluation of this manuscript in the current format, the authors will find in the attached file my considerations/recommendations. 

The authors must improve the quality of the English language. Sometimes, it is difficult to understand some sentences. I indicated some corrections, but the English revision would be recommended. 

Round 2

Reviewer 2 Report

Thank you for your changes. 

I am good With the new manuscript

Reviewer 3 Report

Congratulations. The manuscript was improved. Only minor corrections of the English language, such as in the Discussion section... This results are in agreement

Only minor corrections of the English language, such as in the Discussion section... This results are in agreement